# The binding modes of brazilin and hematein from *Caesalpinia sappan* L. to *Cutibacterium acnes* lipase: Simulation studies

**Maneenuch Pengsawang[1], Borvornwat Toviwek[1], Winyoo Sangthong[2], Apaporn Boonmee[3,4], Phoom Chairatana** 📧[5]*, **Prapasiri Pongprayoon** 📧[1,2]*

**1** Faculty of Science, Department of Chemistry, Kasetsart University, Chatuchak, Bangkok, Thailand, **2** Center for Advanced Studies in Nanotechnology for Chemical, Food and Agricultural Industries, KU Institute for Advanced Studies, Kasetsart University, Bangkok, Thailand, **3** Department of Chemistry, Faculty of Science and Technology, Rambhai Barni Rajabhat University, Chanthaburi, Thailand, **4** Herbal Product for Health and Beauty Research and Development Center, Rambhai Barni Rajabhat University, Chanthaburi, Thailand, **5** Department of Microbiology, Faculty of Medicine Siriraj Hospital, Mahidol University, Bangkok, Thailand

*Co-corresponding authors: phoom.cha@mahidol.ac.th (PC); fsciprpo@ku.ac.th (PP)

## Abstract

The growing concern over antimicrobial resistance in *Cutibacterium acnes* (*C. acnes*) has spurred interest in alternative acne treatments, particularly herbal medicines. This study evaluates the binding affinities of established anti-acne agents—ketoconazole (KET) and tetracycline (TET)—alongside natural compounds, brazilin (BRA) and hematein (HEM), derived from *Caesalpinia sappan* L. (*C. sappan*), to *C. acnes* lipase. Through molecular docking and dynamics simulations, we demonstrate that the asymmetric lipase dimer operates independently. Bulky compounds such as KET and TET inhibit lipase activity via π-π interactions, primarily targeting the lid domain. In contrast, smaller ligands BRA and HEM exhibit unique binding modes: BRA mirrors TET by localizing near the lid domain, while HEM shows dual interactions with both the lid and catalytic sites. These results underscore the potential of BRA and HEM as promising anti-acne agents, indicating that *C. sappan* could be an effective herbal remedy for acne. This study provides a foundation for further exploration of natural products in combating acne and mitigating antimicrobial resistance.

## Introduction

Acne vulgaris is one of the most prevalent dermatological conditions, particularly among adolescents and teenagers worldwide. *Cutibacterium acnes* (*C. acnes*), a Gram-positive anaerobic bacterium, plays a central role in the pathogenesis of acne vulgaris. As a key component of the skin microbiome, *C. acnes* is predominantly found in seborrheic areas where sebaceous glands are highly active. Overgrowth of *C. acnes* is associated with multiple mechanisms contributing to acne development, including follicular hyperkeratinization and alterations in sebum composition, which lead to inflammatory skin responses [1,2]. Notably, *C. acnes* produces enzymes such as lipases and proteases that modify sebum

**Data availability statement:** All relevant data are within the manuscript and its Supporting Information files.

**Funding:** The Office of the National Economic and Social Development Council, and the Office of the Prime Minister through Kasetsart University under the project entitled "Driving Research and Development of Cutting-edge Innovations for ASEAN's Agricultural Leadership". 2. Kasetsart University Research and Development Institute (KURDI) (grant number: FF(KU) 51.67).

**Competing interests:** The authors have declared that no competing interests exist.

**Abbreviations:** BRA, Brazilin; HEM, Hematein; TET, Tetracycline; KET, Ketoconazole; MD, Molecular dynamics.

composition. Lipases, in particular, degrade sebum triglycerides into free fatty acids, compromising the skin barrier, promoting follicular rupture, and triggering inflammation. These alterations increase the likelihood of pore blockage and establish a self-perpetuating cycle of acne development [3].

*C. acnes* lipase plays a pivotal role in this process by hydrolyzing sebum triglycerides into free fatty acids that irritate pilosebaceous follicles, resulting in acne lesions and significant inflammation [4,5]. Studies have demonstrated that reducing free fatty acid production can suppress *C. acnes* growth, highlighting lipase as a critical therapeutic target in acne treatment [6]. Current anti-acne therapies include retinoids and benzoyl peroxide, often combined with antibiotics [7]. However, prolonged antibiotic use has led to the emergence of antimicrobial resistance, underscoring the urgent need for alternative therapeutic approaches. Herbal medicines have gained attention as promising alternatives due to their efficacy and safety. Previous research has demonstrated that certain natural compounds exhibit anti-acne activity through both computational analysis and experimental validation [5,8–10].

Recent structural studies have provided valuable insights into the unique characteristics of *C. acnes* lipase. Unlike most lipases, which are monomeric, *C. acnes* lipase functions as an asymmetric dimer [2]. Seven key interactions (A2-D202, L3-D206, R69-E5, E5-R43, E23-K205, H47-D54, and D54-R294) are essential for its dimerization (Fig 1A). Each subunit comprises two distinct domains: a core domain that houses the catalytic triad (S114, D252, and H285) and a lid domain (residues 144–231), which shields the active site using bulky aromatic residues (F176, F179, W192, and F21; see inset in Fig 1A). While the core domain shares structural similarities with other lipases, the lid domain exhibits unique features that significantly influence its function [2].

Furthermore, three crystal structures of *C. acnes* lipase, corresponding to the open (PDB code: 6KHM), closed (PDB code: 6KHK), and blocked (PDB code: 6KHL) conformations, have been reported [2]. Superimposition of these X-ray structures revealed a conserved backbone scaffold (S1 Fig). Variations among these structures are attributed to ligand binding, which induces sidechain reorientations and modifies pocket cavities. Such structural insights are crucial for understanding the mechanisms of lipase inhibition.

Several drugs and natural compounds, including tetracycline (TET) [4], erythromycin [4], fospirate [4], and ketoconazole (KET) [8], inhibit *C. acnes* lipase. Among these, tetracycline and ketoconazole have emerged as particularly promising inhibitors [4–8] (Fig 1B). Recent studies have identified *Caesalpinia sappan* L. (*C. sappan*; Leguminosae) as a natural source of anti-acne compounds [11,12]. Brazilin (BRA), a homoisoflavonoid dye primarily derived from the heartwood of *C. sappan*, exhibits notable antibacterial and anti-inflammatory properties [13], making it a potential candidate for acne treatment [14,15]. BRA has also been reported to inhibit *C. acnes* lipase activity, reducing the breakdown of sebum triglycerides, a key factor in acne pathogenesis [16]. Experimental studies have demonstrated that BRA significantly suppresses *C. acnes* growth, with a minimum inhibitory concentration (MIC) of 15.6 μg/mL [13]. Its mechanism of action involves interaction with the active site of *C. acnes* lipase, suggesting dual therapeutic roles in inhibiting bacterial proliferation and modulating inflammatory responses associated with acne lesions [17]. Another compound, hematein (HEM), has shown similar potential by influencing lipid metabolism, particularly in the context of *C. acnes* lipase activity [18]. Despite experimental evidence supporting the inhibitory effects of BRA on *C. acnes* lipase, its precise binding mechanism remains unclear.

This study investigates the anti-acne properties of BRA and HEM in comparison with KET and TET. Using molecular dynamics (MD) simulations, we elucidate the binding mechanisms of these compounds with *C. acnes* lipase, providing insights that may pave the way for future therapeutic applications of *C. sappan* in the treatment of acne vulgaris.

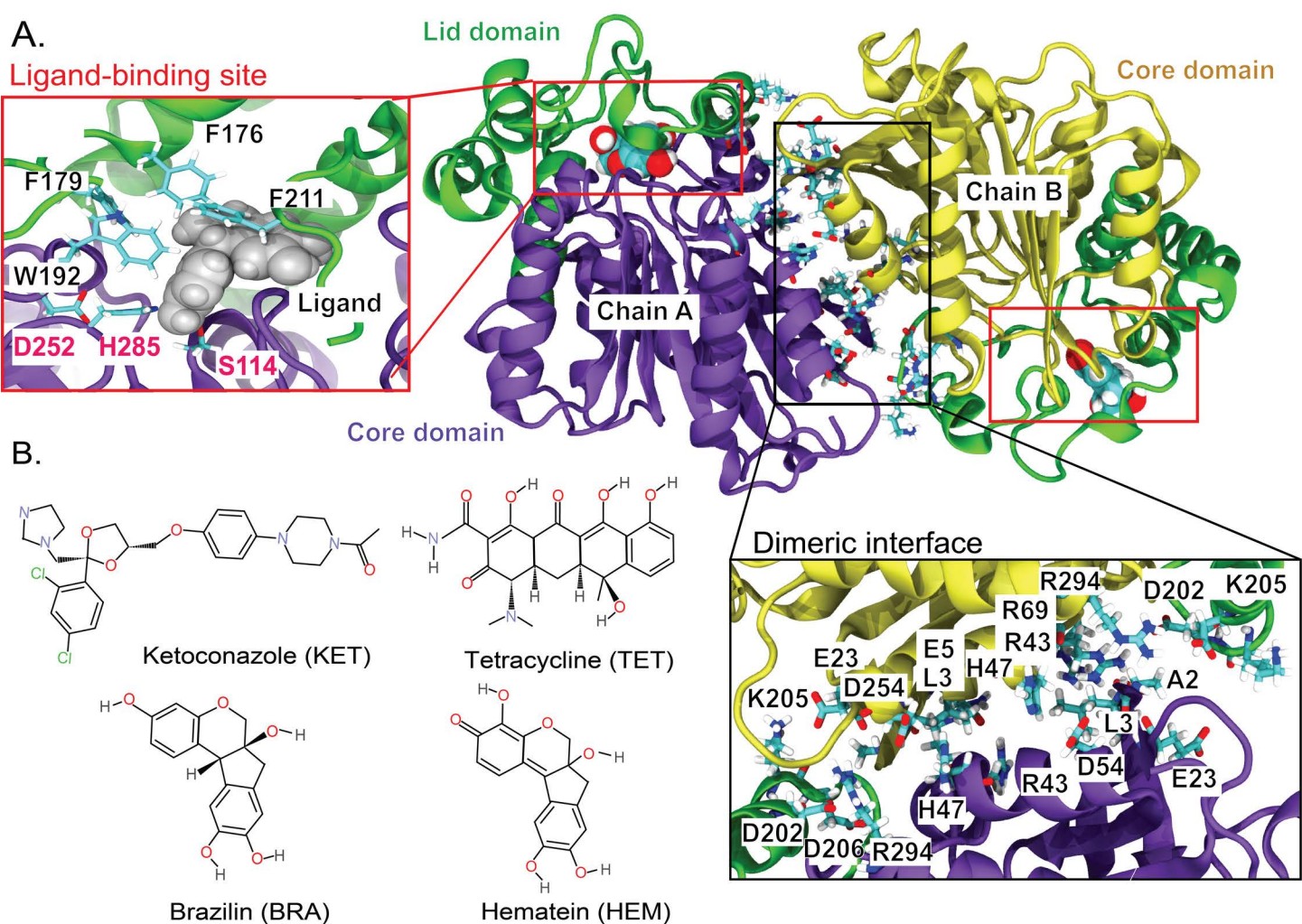

**Fig 1. (A) The dimeric structure of C. acnes lipase, with chain A (purple) and chain B (yellow), highlighting the lid domain.** Each monomer contains a single ligand-binding site, as indicated by the red insets. The catalytic triad (S114, D252, and H285, shown in red) and hydrophobic residues (F176, F179, F211, and W192, shown in black) are depicted. Residues forming the interface between the two subunits are displayed in the black inset. (B) Chemical structures of the ligands used in this study: ketoconazole (KET), tetracycline (TET), brazilin (BRA), and hematein (HEM).

## Materials and methods

### Preparation of lipase complexed with brazilin (BRA), hematein (HEM), ketoconazole (KET), and tetracycline (TET)

The crystal structure of the open form of lipase from *C. acnes* was obtained from the RCSB Protein Data Bank (PDB ID: 6KHM; resolution: 2.40 Å). Missing residues were modeled using Modeller 10.4 [19], and the protonation states of charged residues were adjusted to physiological pH. The chemical structures of BRA, HEM, KET, and TET were retrieved from the PubChem database [21–23], and ligand topologies were generated using ACPYPE with the AMBER force field [24]. Molecular docking was performed using GOLD 5.3 software [25] with default parameters for flexible ligand docking to obtain the ligand-lipase complexes as defined within the program. The binding site was defined as all protein residues within 1 nm from the center of the active site. The ChemPLP and ChemScore fitness functions available in

GOLD were used [20]. The ligand-lipase complex with the highest Goldscore was selected as the initial structure for subsequent simulation studies (S1 Table). Each protein-ligand complex was then placed in a cubic simulation box (dimension of 10x10x10 nm$^3$) and solvated with TIP3P water molecules. A dimer has the total charges of -30, thus 30 Na$^+$ ions were added as counter ions to neutralize the system, then 0.15 M NaCl was introduced into the system to mimic physiological conditions (pH 7).

## Simulation protocols

All simulations were performed using the GROMACS v.2020 software package (http://www.gromacs.org/) [26] with the AMBER99SB-ILDN force field [27]. Energy minimization was carried out using the steepest descent algorithm until the maximum force between atoms was below 1,000 kJ/mol·nm to reduce steric conflicts. Long-range electrostatic interactions were computed via the Particle Mesh Ewald (PME) method [28], employing a short-range cutoff of 1 nm, a Fourier spacing of 0.12 nm, and fourth-order spline interpolation. The temperature of the lipase, substrate, solvent, and ions was individually coupled using the v-rescale thermostat [29], maintaining a temperature of 300 K with a coupling constant ($\tau_t$) of 0.1 ps. The Parrinello-Rahman barostat was used to regulate pressure at 1 bar with a coupling constant ($\tau_p$) of 1 ps. A 2-fs integration time step was applied, and coordinates were saved every 2 ps for subsequent analysis. The Periodic Boundary Condition (PBC) was applied in all directions. Each system underwent a 10-ns equilibration phase, followed by a 500-ns production run. After the course of equilibration, all systems became equilibrated as seen in S2 Fig. Simulations were conducted in duplicates (the suffixes of "1 and 2" are used to refer to simulation 1 and simulation 2). The suffixes "A" and "B" denote chains A and B, respectively.

The results presented in this study are averages derived from two independent simulations. Data analysis was carried out using GROMACS and in-house codes. Visualization of the molecular dynamics was performed using Visual Molecular Dynamics (VMD) software [30]. The initial structure from each run was used as a reference for calculating C-alpha root-mean-square deviation (RMSD), and root-mean-square fluctuation (RMSF). Principal Component Analysis (PCA) was conducted using the "gmx covar" and "gmx anaeig" functions in GROMACS. Hydrogen bond counts were calculated with the "gmx hbond" function, utilizing the default setting: a hydrogen-donor–acceptor cutoff angle of 30° and a cutoff radius (X-acceptor) of 0.35 nm. Binding free energies of the protein-ligand complexes were determined using MM/PBSA, executed via the "gmx mmpbsa" tool. The molecular electrical potential (MEP) surfaces of all ligands were computed by Gaussian 16 package [21] using Density Functional Theory (DFT) with B3LYP6.31G(d,p) basis set. The GaussView 5 was used for MEP visualization [22]. The color range of electrostatic potential surface (ESP) was chosen from red (-0.05 a.u.) to blue (+0.05 a.u.).

## Results and discussion

In this study, the binding mechanisms of active compounds (brazilin (BRA) and hematein (HEM)) from *C. sappan* to lipase from *C. acne* were investigated in comparison to ketoconazole (KET) and tetracycline (TET), both of which were reported as lipase inhibitors [8,23]. As depicted in Fig 2, Root-Mean-Square Deviations (RMSDs) and Root-Mean-Square Fluctuations (RMSFs) were calculated to analyze the protein dynamics, with the initial structure at 0 ns used as a reference. Notably, the binding of BRA and HEM appears to increase the protein flexibility (Fig 2A). Chains A and B of the lipase in the presence of BRA and HEM exhibit greater structural flexibility compared to KET and TET (Fig 2A). The RMSFs presented in Fig 2B highlight the regions of high flexibility at residues 5-20, 65-105, and 142-205

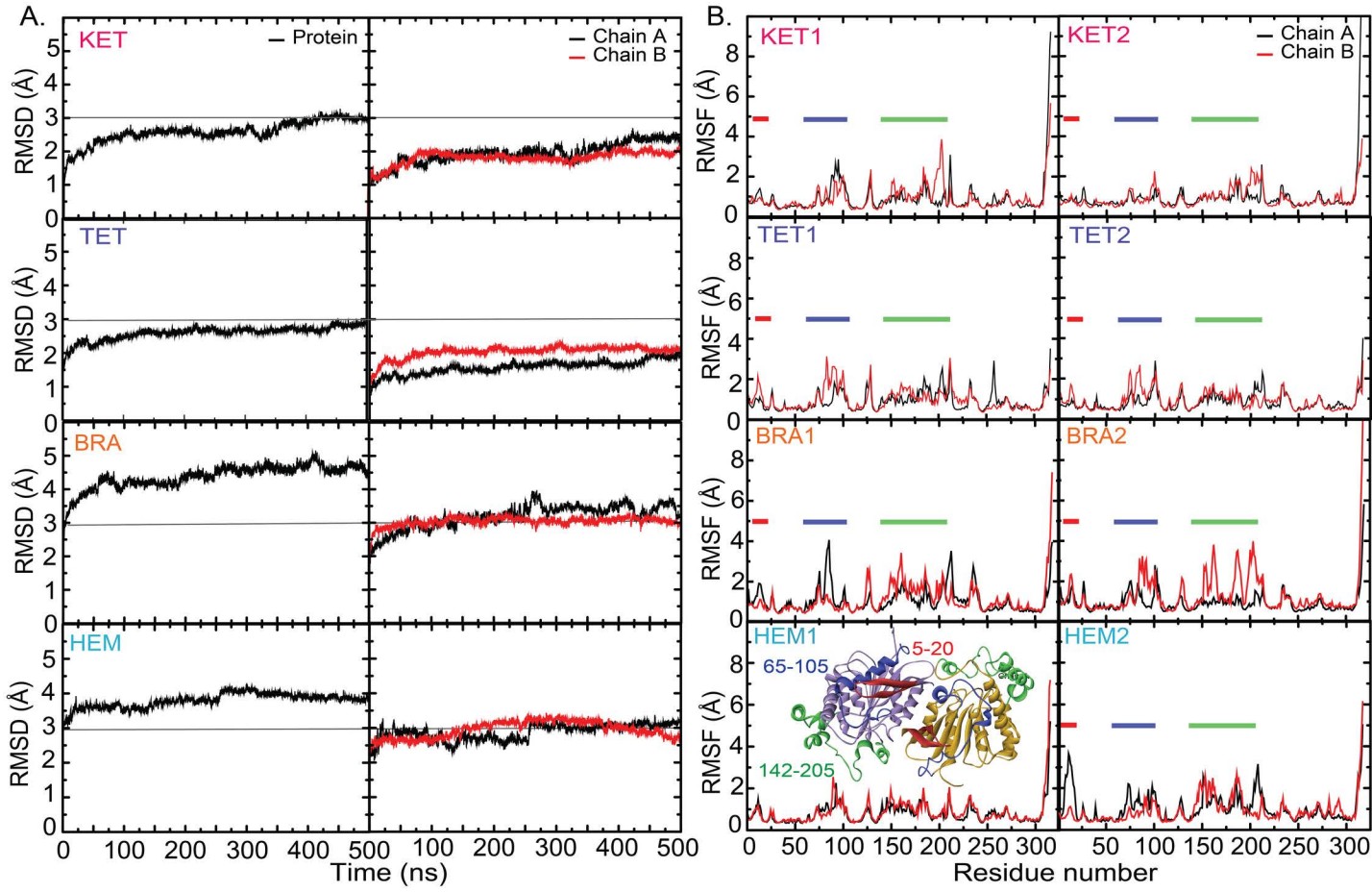

**Fig 2. C-alpha RMSDs of the whole protein and each chain (A) and RMSFs (B).** For RMSFs, red, blue, and green bands stand for the high fluctuated regions where their locations are shown in an inset.

(Fig 2B), where residues 142-205 are part of the lid domain (Fig 1A). In addition, two subunits show different degrees of structural flexibility and fluctuation demonstrating the independent dynamics between two chains. The PCA analysis can confirm the high flexibility of lid domain in BRA and HEM (S1B Fig).

To further investigate the ligand-binding mechanisms, the number of ligand-protein and ligand-water hydrogen bonds was calculated (Fig 3). As shown in Fig 3, HEM and BRA engage in a similar number of water interactions as KET and TET drugs; however, both compounds appear to form a more consistent number of water contacts (Fig 3A). In the case of protein-ligand hydrogen bonds, BRA and TET form a similar number of hydrogen bonds to the existing drug TET (~2 permanent hydrogen bonds) (Fig 3B). In contrast, the smaller number of hydrogen bonds in KET indicates that hydrogen bonding is not the primary driving force for KET binding (Fig 3B). Seemingly, the binding of KET is driven by hydrophobic interactions. This suggestion is further supported by the electrostatic potential shown in Fig 3C. The bulky KET exhibits increased hydrophobicity compared to the other compounds. These findings suggest that KET binding is primarily driven by non-polar interactions.

The binding energies between ligands and protein (Table 1) confirm that the hydrophobic interactions are the major driving force for ligand binding. KET and TET display significantly

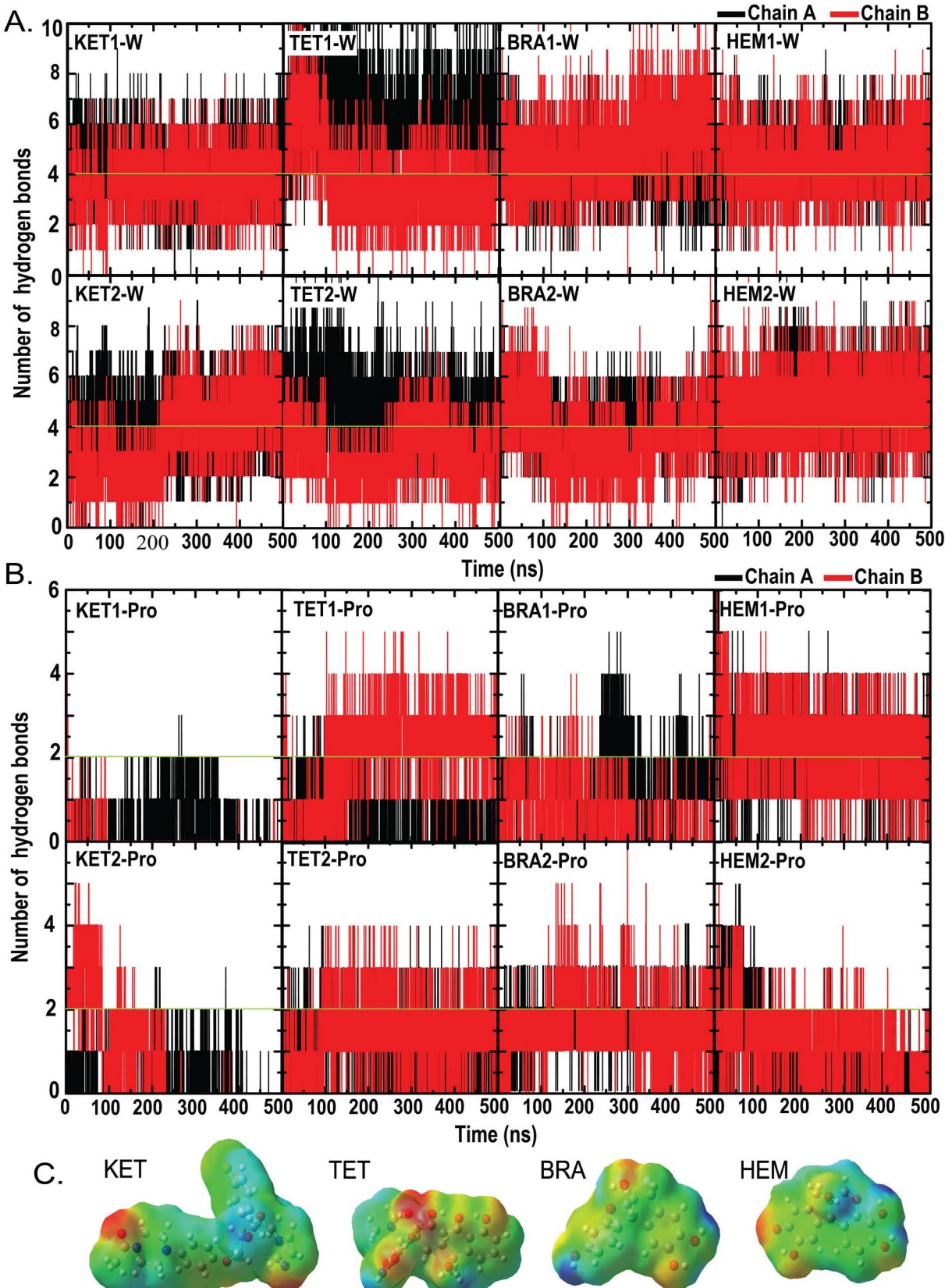

**Fig 3. The number of hydrogen bonds formed by each ligand with the protein (A) and water (B) in chain A (black) and B (red).** (C) Electrostatic potential map of each ligand, with electronegative and positive regions shown in red and blue, respectively.

**Table 1. Interaction energies (kJ/mol) between each ligand (KET, TET, BRA, and HEM) and *C. acnes* lipase. The data after 350 ns were used to calculate the interaction energies.**

| System | Chain A (kJ/mol) | | | Chain B (kJ/mol) | | |
|---|---|---|---|---|---|---|
| | $\Delta E_{vdW}$ | $\Delta E_{Elec}$ | Total Energy | $\Delta E_{vdW}$ | $\Delta E_{Elec}$ | Total Energy |
| KET1 | -184.87 ± 19.90 | -40.13 ± 22.53 | -225.00 ± 32.97 | -230.43 ± 13.65 | -49.85 ± 15.41 | -280.28 ± 22.49 |
| KET2 | -194.99 ± 15.88 | -25.22 ± 26.40 | -220.21 ± 31.30 | -168.56 ± 21.07 | -40.87 ± 26.20 | -209.43 ± 37.83 |
| TET1 | -174.28 ± 15.21 | -56.03 ± 24.70 | -230.32 ± 30.99 | -214.53 ± 15.17 | -29.10 ± 19.63 | -243.63 ± 27.53 |
| TET2 | -201.01 ± 10.43 | -47.23 ± 14.21 | -248.24 ± 16.88 | -222.79 ± 12.84 | -54.55 ± 18.57 | -227.34 ± 22.62 |
| BRA1 | -151.82 ± 19.67 | -67.76 ± 17.76 | -219.58 ± 26.02 | -120.59 ± 12.26 | -16.24 ± 29.19 | -139.58 ± 27.32 |
| BRA2 | -161.02 ± 12.29 | -73.05 ± 12.37 | -232.99 ± 15.79 | -130.65 ± 21.85 | -57.64 ± 41.25 | -188.29 ± 41.14 |
| HEM1 | -160.58 ± 10.54 | -100.24 ± 17.77 | -260.82 ± 19.99 | -139.21 ± 10.80 | -98.83 ± 24.75 | -238.04 ± 27.08 |
| HEM2 | -155.54 ± 9.99 | -44.41 ± 16.88 | -199.96 ± 19.83 | -134.57 ± 9.83 | -42.45 ± 17.76 | -177.03 ± 18.55 |

high van der Waals energies (~170-230 kJ/mol), whereas BRA and HEM exhibit lower hydrophobic forces (~120-150 kJ/mol) (Table 1). In contrast, BRA and HEM induce more substantial electrostatic energies (up to 100 kJ/mol) due to the higher number of hydrogen bonds found in Fig 3. The bulkier KET and TET appear to show hydrophobic-dominating features, whereas BRA and HEM are more hydrophilic. Our findings are consistent with previous studies, which indicate that KET primarily relies on hydrophobic interactions for protein binding [24,25]. Effective inhibitors for *C. acne* lipase appear to require large hydrophobic moieties. When considering total binding energies, all ligands exhibit comparable binding abilities for lipase, with KET and TET showing slightly stronger binding capabilities (Table 1). Both lipase monomers have similar capacities to bind KET and TET, but chain A shows a higher affinity for the smaller BRA and HEM compared to chain B. This finding highlights the influence of the asymmetric dimer on the binding of smaller ligands.

To investigate how the ligands align within the binding pocket, the superimpositions of each ligand over time are depicted in Fig 4, with surrounding residues labeled. It is evident that KET and TET undergo significant displacement within the pocket, particularly KET, while the smaller ligands, BRA and HEM, exhibit less mobility. The considerable reorientation of KET is corroborated by the high RMSD values (S3 Fig). This high flexibility of large inhibitors within the binding pocket is consistent with findings from other inhibitor-lipase studies [26–30].

Among the studied ligands, HEM not only forms the highest number of protein contacts, but also demonstrates greater water accessibility (Figs 3 and 4). Notably, no consistent ligand configuration is observed between the two chains. With the exception of HEM, the ligands appear to adopt distinct binding environments. This can be attributed to the asymmetric nature of the dimer, which induces different lid conformations in each chain, resulting in distinct ligand-binding conformations (Fig 4). The displacement of ligands in most cases reflects the large and flexible nature of the binding cavity. KET, TET, and BRA seem to be primarily located in the lid domain, but the tail regions of TET1B and TET2B appear to extend toward the catalytic site (H285) (Fig 4).

In contrast, HEM exhibits a different binding mechanism. All HEM molecules remain close to the catalytic site (S114 and H285), despite some structural fluctuations (Fig 4). HEM forms multiple interactions with the residues lining the catalytic site, resulting in tighter packing and more stable binding in this site.

Previous X-ray studies revealed that substrates are trapped by aromatic residues in the lid domain (F176, F179, W192, and F21, where F176 and F211 are identified as gating residues) [2] (Fig 1A). To understand how the lid domain facilitates the ligand binding, the Cα-Cα

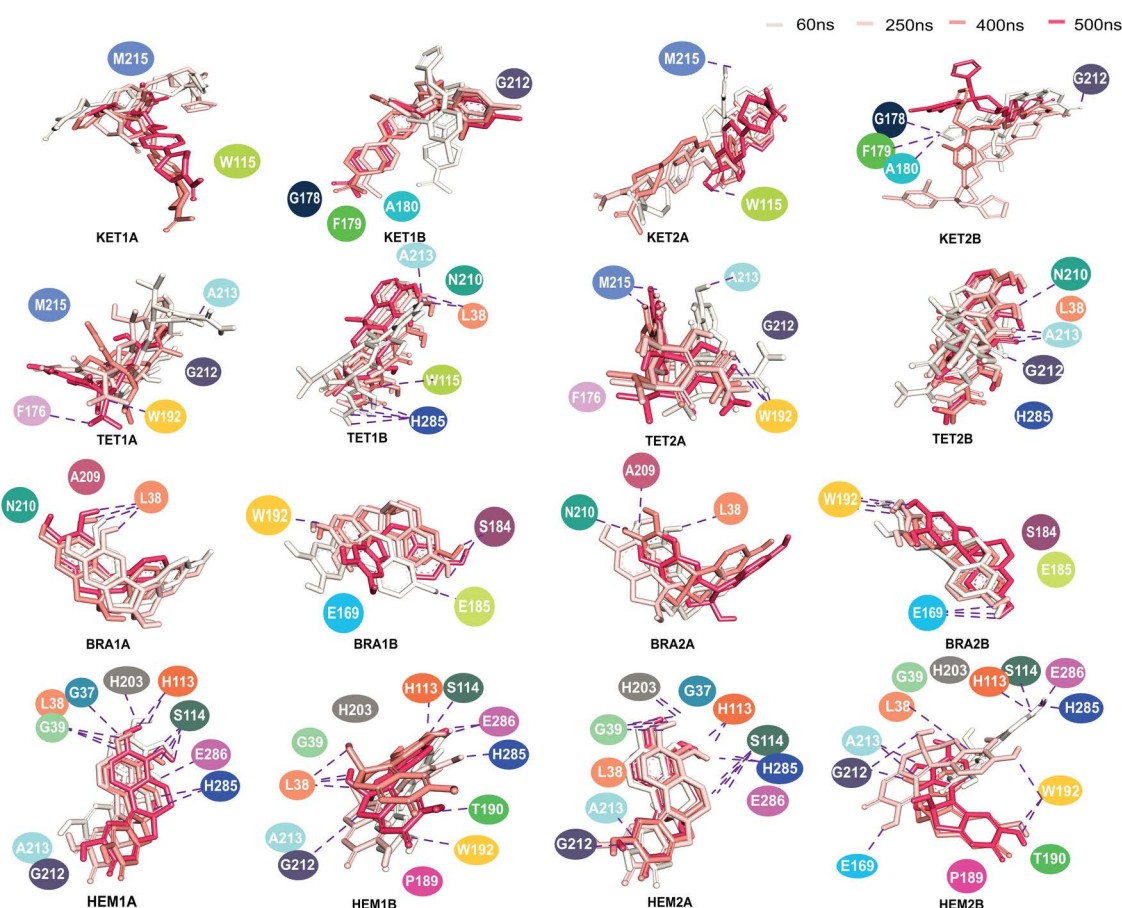

**Fig 4. Superimposition of each ligand at 60ns, 250ns, 410ns, and 500ns, respectively.** Residues within 0.35 nm of each ligand are also shown. The dashed line indicates the presence of hydrogen bonds.

distances between the gating residues (F176 and F211) were calculated (Fig 5A). The observed differences in F176-F211 distances between the two subunits clearly confirm the asymmetric and independent functioning of each chain (Fig 5A and S4 Fig).

Comparisons with crystal structures [2], F176-F211 distances of 1.55-1.57 nm are observed in the closed and blocked forms of lipase (PDB code: 6KHK and 6KHL), while a longer distance of 1.62 nm is identified in the open form (PDB code: 6KHM). A shorter F176-F211 distance ($\leq 1.5$ nm) is found when the ligand is sandwiched between aromatic residues in the lid domain (conformation 1 in Fig 5B and S5 Fig). In contrast, a wider F176-F211 distance ($\geq 1.5$ nm) is found when ligands align perpendicularly to the protein axis (conformation 2 in Fig 5B and S4 Fig).

Despite being retained within the pocket, lipase cannot fully accommodate the bulky structures of KET and TET (Fig 5C). Particularly in KET1B and KET2B, KET appears to be vertically pinched by the lid-opening residues (F176 and F211), with half of the KET molecule exposed to the solvent (S4 Fig), leading to the shortening of F176-F211 distance (Fig 5A). This ligand-trapping orientation is reported here for the first time and is also observed with TET (S5 Fig). Previous studies displayed that bulky inhibitors primarily align in the lid domain of other lipases [26–29], consistent with the binding behavior of KET and TET in this study.

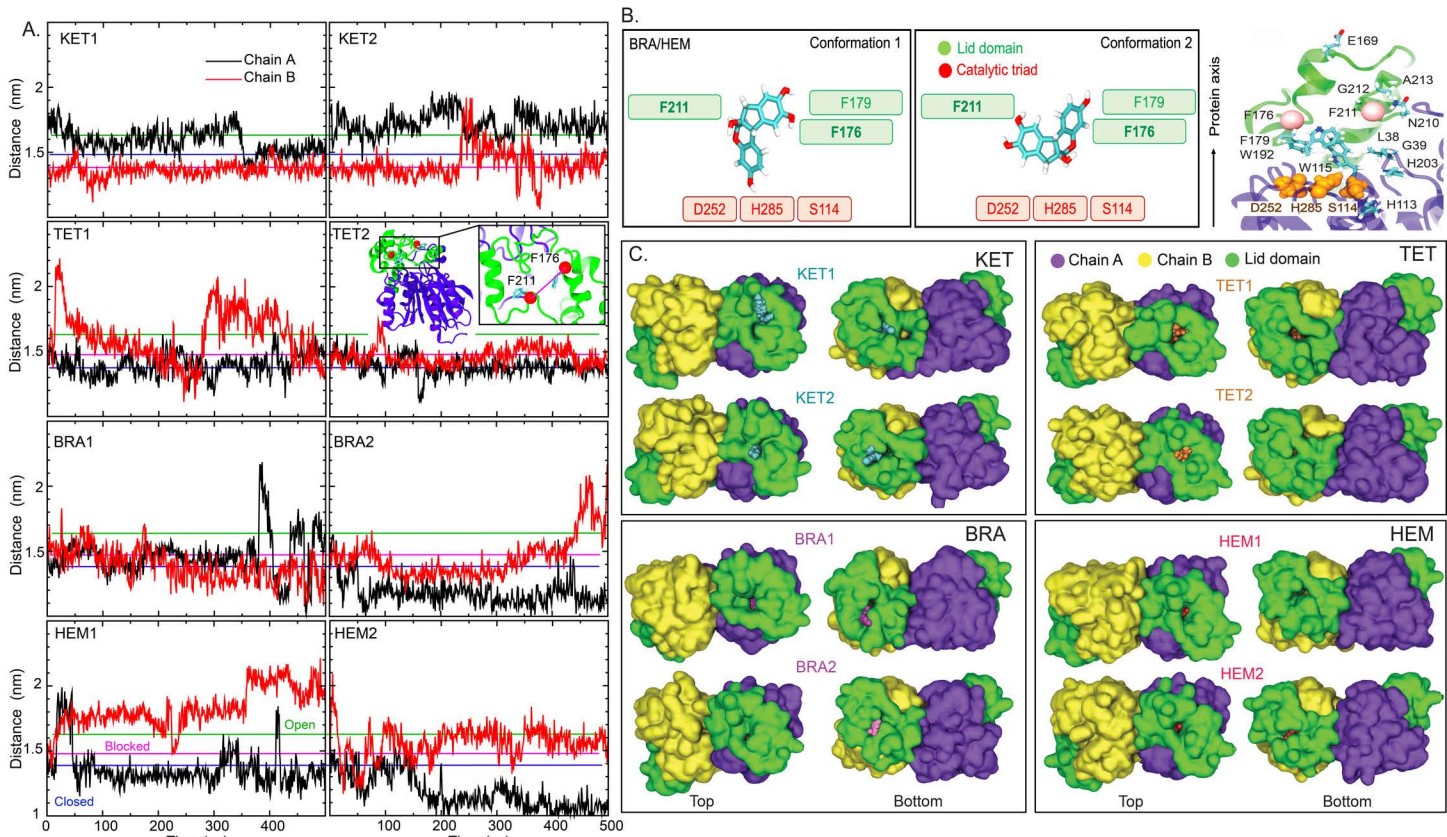

**Fig 5. (A) Distances between the C-alpha atoms of the gating residues (F179 and F211) in all systems, with their locations shown in the inset.** The green and blue lines represent the Cα-Cα distances between F176 and F211 measured from the open form (1.62 nm) (PDB code: 6KHM) and the closed (1.39 nm) or blocked (1.47 nm) forms (PDB code: 6KHK and 6KHL) of the *C. acnes* lipase structures. (B) Two orientations of the ligands inside the binding pocket, with the aromatic gating residues (F176, F179, and F211) and the catalytic triad (S114, D252, and H285) labeled. A side view of lipase, highlighting key residues involved in ligand binding and the catalytic triad (depicted as orange VdW surfaces), is shown on the right. (C) Top and bottom views of the lipase surfaces for each system, with chains A and B colored violet and yellow, respectively, and the lid domain shown in green. The ligands KET, TET, BRA, and HEM in the active site are depicted in cyan, orange, magenta, and red, respectively.

In contrast, the smaller BRA and HEM fit tightly within the pocket. Their binding can result in both blocked and closed conformations of the lipase structures (Fig 5C). Further discussion on this topic is presented later in the text.

To gain a deeper understanding of the lipase-ligand interaction network, the number of hydrogen bonds between each ligand and the protein was examined (Fig 6). As discussed earlier, the binding of KET is primarily driven by dispersion forces, which explains the low number of hydrogen bonds observed (Fig 6). KET does not form significant hydrogen bonds with the catalytic residues (S114, D252, and H285), with only a transient interaction with S114 observed (Fig 6A). Instead, KET tends to form hydrogen bonds with residues T190 and W192 in KET1, and with G212 and A213 in KET2, located in the lid domain (Fig 6A).

For the more polar ligand TET, additional interactions are observed with L38, S114, W115, E169, N210, F211, and H285 (Fig 6A). Although both KET and TET primarily align near the lid domains, the interactions of TET with S114 and H285 suggest that it penetrates deeper into the active site.

The smaller ligands, BRA and HEM, exhibit distinct binding modes. BRA forms an interaction network similar to that of KET and TET, primarily localizing near the lid domain without

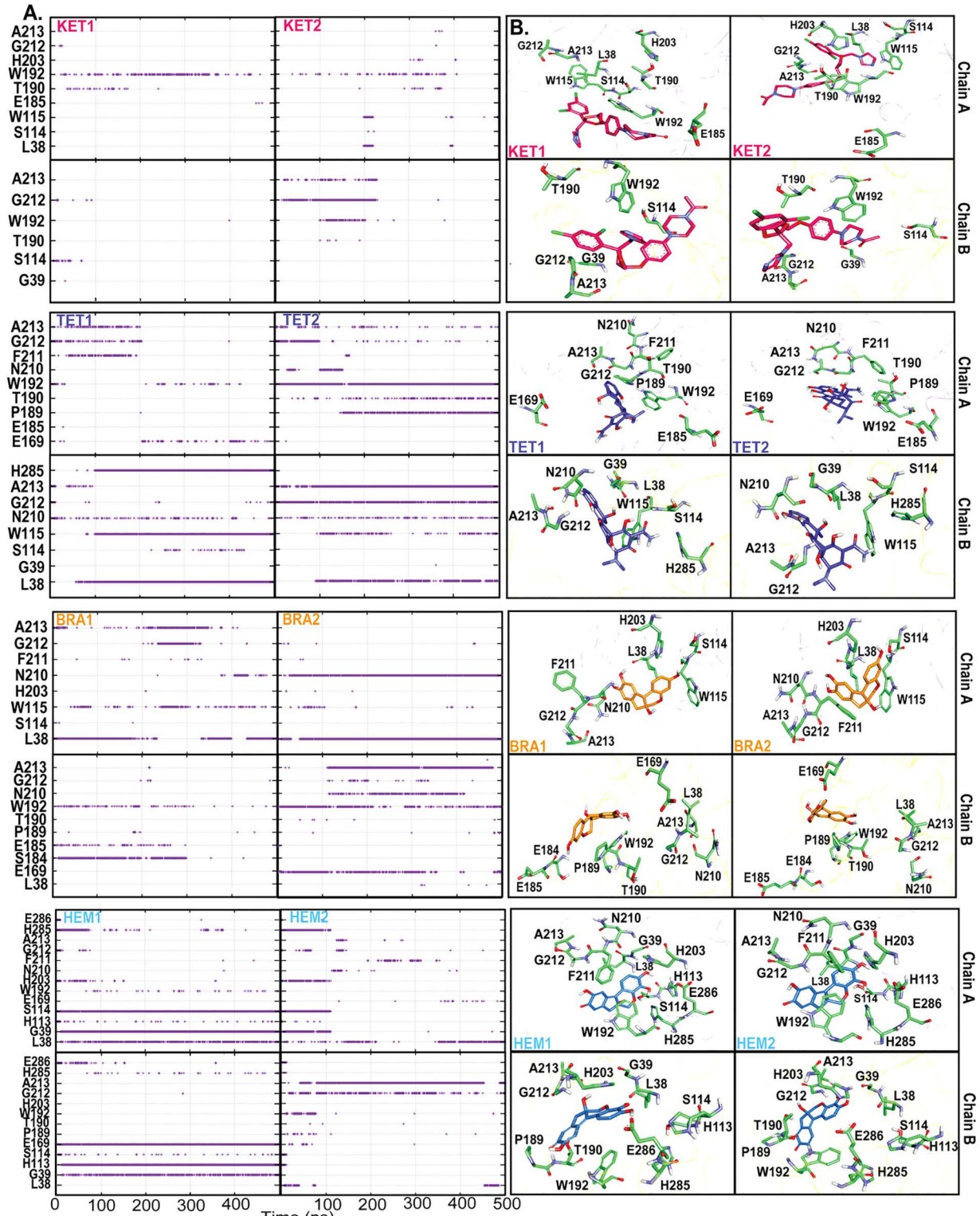

**Fig 6. (A)** Occurrence of hydrogen bonds as a function of time. **(B)** Orientations of each ligand in the binding pocket where key residues are labeled.

interacting with the catalytic triad (Fig 6A). In contrast, HEM packs more closely to the catalytic triad. While BRA appears to block the entrance to the binding pocket (lid domain), HEM occupies both the catalytic pocket and the lid domain.

In the HEM1 conformation, HEM is oriented upright and interacts with both the catalytic triad and lipophilic moieties (lid domain). It forms hydrogen bonds with G39, H113, E169, and the catalytic residues S114 and H285 (Fig 6 and conformation 2 in S5 Fig). For HEM2, it seems to translocate from the catalytic site toward the lid domain, forming hydrogen bonds with G212 and A213 (Fig 6 and conformation 1 in S5 Fig).

Overall, KET, TET, and BRA are primarily localized near the lid domain, while only HEM can penetrate the catalytic region. Most proposed lipase inhibitors were computationally identified to bind to the lid domain through $\pi$ - $\pi$ stacking interactions [2,30,31]. In contrast to the other ligands, HEM exhibits a dual mode of action; it can both obstruct the lid domain and engage with the catalytic triad, whereas KET, TET, and BRA predominantly induce a blocked conformation of the lipase.

Our findings suggest the potential application of BRA and HEM from *C. sappan* as an alternative herbal medicine for acne treatment. Although previous studies have identified BRA as an anti-acne agent through the inhibition of lipase activity [16,32], detailed mechanistic insight remain lacking. Notably, there are no reported studies on the anti-acne activity of HEM. This study explores the potential of HEM to inhibit *C. acne* lipase and further experiments are required to confirm our findings.

## Conclusion

Conventional acne treatment with antibiotics raises concerns due to the development of antimicrobial resistance in *C. acnes*. Consequently, there has been a growing interest in alternative strategies, such as herbal medicine, to minimize the increase in antimicrobial resistance and mitigate severe side effects. This study examines the binding affinities of existing anti-acne agents (KET and TET) and natural products (BRA and HEM) derived from *C. sappan* in relation to *C. acne* lipase. Our findings indicate that the asymmetric lipase dimer operates independently. Bulky compounds, in particular, effectively inhibit lipase by forming $\pi-\pi$ interactions with the lid domain [30], consistent with previous studies. Specifically, KET is constricted within the lid domain and relies on the hydrophobic interactions for binding, whereas TET interacts with the lipase through both hydrogen bonds and hydrophobic forces, remaining predominantly trapped at the lid domain.

In contrast, the smaller BRA and HEM from *C. sappan*, exhibit differing modes of action. Although both BRA and HEM bind to the lipase via electrostatic and van der Waal forces, BRA behaves similarly to TET by localizing near the lid domain. Conversely, HEM can interact with both the lid domain and the catalytic site. These results indicate that both BRA and HEM exhibit promising affinities for *C. acne* lipase, suggesting that they contribute to the anti-acne activity of *C. sappan*. Notably, the binding efficacy of HEM positions it as a potential anti-acne agent. This work underscores the potential of *C. sappan* as a therapeutic option for acne treatment.

## Supporting information

**S1 Fig. Superimposition of three lipase enzymes obtained from PDB databank (PDB codes: 6KHM, 6KHL, and 6KHK).**
(TIF)

**S2 Fig. Energies (kJ/mol), Temperature (K), and Pressure (bar) of all systems in the equilibration step.**
(TIF)

**S3 Fig. RMSDs of KET, TET, BRA, and HEM in each chain in all systems.**
(TIF)

**S4 Fig. Top views of superimpositions of chains A and B with ligand in each system. Chains A and B are colored in violet and yellow.**
(TIF)

**S5 Fig. Cartoon views of ligand orientation at the gate (F176 and F211) when the distances between C-alpha atoms of F176 and F211 are < 1.5 nm and > 1.5 nm.**
(TIF)

**S1 Table. Goldscores of ketoconazole (KET), tetracycline (TET), brazilin (BRA), and hematein (HEM) with lipase enzyme.**
(XLSX)

## Acknowledgment

We would like to thank the Kasetsart University HPC center (Nontri-AI) for computer support.

## Author contributions

**Conceptualization:** Phoom Chairatana, Prapasiri Pongprayoon.

**Data curation:** Maneenuch Pengsawang, Prapasiri Pongprayoon.

**Formal analysis:** Maneenuch Pengsawang, Borvornwat Toviwek, Winyoo Sangthong.

**Funding acquisition:** Prapasiri Pongprayoon.

**Investigation:** Maneenuch Pengsawang, Borvornwat Toviwek, Winyoo Sangthong.

**Methodology:** Maneenuch Pengsawang, Borvornwat Toviwek, Winyoo Sangthong, Apaporn Boonmee, Phoom Chairatana, Prapasiri Pongprayoon.

**Resources:** Apaporn Boonmee.

**Supervision:** Phoom Chairatana, Prapasiri Pongprayoon.

**Validation:** Maneenuch Pengsawang, Phoom Chairatana, Prapasiri Pongprayoon.

**Visualization:** Maneenuch Pengsawang, Phoom Chairatana, Prapasiri Pongprayoon.

**Writing – original draft:** Maneenuch Pengsawang, Phoom Chairatana, Prapasiri Pongprayoon.

**Writing – review & editing:** Maneenuch Pengsawang, Phoom Chairatana, Prapasiri Pongprayoon.

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
