## [Decision Letter · Decision Letter 0]

21 Nov 2024

PONE-D-24-45122The binding modes of brazilin and hematein from Caesalpinia sappan L.  to Cutibacterium acnes lipase: Simulation studies.PLOS ONE

Dear Dr. Chairatana,

Thank you for submitting your manuscript to PLOS ONE. After careful consideration, we feel that it has merit but does not fully meet PLOS ONE’s publication criteria as it currently stands. Therefore, we invite you to submit a revised version of the manuscript that addresses the points raised during the review process.

Please revise the manuscript addressing the concerns of all three reviewers. Reviwer #2 was critical, however, I believe his/her concerns can be addresses during the revision.

We look forward to receiving your revised manuscript.

Kind regards,

Kshatresh Dutta Dubey

Academic Editor

PLOS ONE

Journal Requirements: When submitting your revision, we need you to address these additional requirements. 1. Please ensure that your manuscript meets PLOS ONE's style requirements, including those for file naming. The PLOS ONE style templates can be found at https://journals.plos.org/plosone/s/file?id=wjVg/PLOSOne_formatting_sample_main_body.pdf and https://journals.plos.org/plosone/s/file?id=ba62/PLOSOne_formatting_sample_title_authors_affiliations.pdf 2. Please note that PLOS ONE has specific guidelines on code sharing for submissions in which author-generated code underpins the findings in the manuscript. In these cases, we expect all author-generated code to be made available without restrictions upon publication of the work. Please review our guidelines at https://journals.plos.org/plosone/s/materials-and-software-sharing#loc-sharing-code and ensure that your code is shared in a way that follows best practice and facilitates reproducibility and reuse. 3. Thank you for stating the following financial disclosure: "Kasetsart University Research and Development Institute (Grant no. FF(KU)51.67)  Driving Research and Development of Cutting-edge Innovations for ASEAN's Agricultural Leadership by the Office of the National Economic and Social Development Council, and the Office of the Prime Minister " Please state what role the funders took in the study.  If the funders had no role, please state: ""The funders had no role in study design, data collection and analysis, decision to publish, or preparation of the manuscript."" If this statement is not correct you must amend it as needed. Please include this amended Role of Funder statement in your cover letter; we will change the online submission form on your behalf. 4. Thank you for stating the following in the Acknowledgments Section of your manuscript: "We would like to would like to thank Kasetsart University Research and Development Institute (Grant no. FF(KU)51.67) and the Office of the National Economic and Social Development Council, and the Office of the Prime Minister through Kasetsart University under the project entitled "Driving Research and Development of Cutting-edge Innovations for ASEAN's Agricultural Leadership" for financial support. We also thank the Kasetsart University HPC center (Nontri-AI) for computer support." We note that you have provided funding information that is not currently declared in your Funding Statement. However, funding information should not appear in the Acknowledgments section or other areas of your manuscript. We will only publish funding information present in the Funding Statement section of the online submission form. Please remove any funding-related text from the manuscript and let us know how you would like to update your Funding Statement. Currently, your Funding Statement reads as follows: "Kasetsart University Research and Development Institute (Grant no. FF(KU)51.67)  Driving Research and Development of Cutting-edge Innovations for ASEAN's Agricultural Leadership by the Office of the National Economic and Social Development Council, and the Office of the Prime Minister " Please include your amended statements within your cover letter; we will change the online submission form on your behalf. 5. Please include captions for your Supporting Information files at the end of your manuscript, and update any in-text citations to match accordingly. Please see our Supporting Information guidelines for more information: http://journals.plos.org/plosone/s/supporting-information. 6. Please review your reference list to ensure that it is complete and correct. If you have cited papers that have been retracted, please include the rationale for doing so in the manuscript text, or remove these references and replace them with relevant current references. Any changes to the reference list should be mentioned in the rebuttal letter that accompanies your revised manuscript. If you need to cite a retracted article, indicate the article’s retracted status in the References list and also include a citation and full reference for the retraction notice.

Reviewers' comments:

Reviewer's Responses to Questions

**Comments to the Author**

1. Is the manuscript technically sound, and do the data support the conclusions?

Reviewer #1: Yes

Reviewer #2: No

Reviewer #3: Yes

2. Has the statistical analysis been performed appropriately and rigorously? 

Reviewer #1: Yes

Reviewer #2: N/A

Reviewer #3: Yes

3. Have the authors made all data underlying the findings in their manuscript fully available?

Reviewer #1: Yes

Reviewer #2: No

Reviewer #3: Yes

4. Is the manuscript presented in an intelligible fashion and written in standard English?

Reviewer #1: No

Reviewer #2: Yes

Reviewer #3: Yes

5. Review Comments to the Author

Reviewer #1: In reviewing the manuscript, I have noted several instances of spelling errors that should be addressed before publication. These minor issues, while not significantly affecting the overall content, could impact clarity of the manuscript. A careful proofreading of the document is recommended to correct these errors.

Reviewer #2: There are several technical queries with the manuscript mentioned below, which therefore do not allow this manuscript for publishing in its current state.

1. Pg. 13, line 1; default docking protocol; what are those default protocols, it should be mentioned somewhere in the literature (or in the SI).

2. Is “1x1x1 nm3” the size of the box or padding distance, if it’s the size of the box as mentioned, then mention the padding distance?

3. Whether PBC effects were taken into consideration or not?

4. What is the charge of the protein? How many counter ions were added?

5. At what pH was the protonation state of the residues (specially H) decided?

6. How were the ligands parametrized? Which force field, which tool?

7. During energy minimizations was the water relaxed before minimizing the entire system?

8. Since the protocol seems incomplete, can the consistent plateau of minimized energies be provided by the authors in the SI, alongside the equilibrated pressures and temperatures as well?

9. Pg. 15, line 5, the authors say that “This observation is further supported by the electrostatic potential shown in Figure 3C.” but no where in the manuscript is mentioned how the ESPs are generated?

10. Also corroborating the H-bonding observed in dynamics with ESPs does not go well because one is based on force field while the other is quantum mechanical? So, this corroboration must be omitted or must be explained in some other way? Although the authors may take snapshots from the trajectory and perform QM to justify their findings.

11. Why in the figure 4 the superimposition of each ligand at 60ns, 250ns, 410ns, and 500ns, are shown; instead of representing them at consistent intervals. It is like choosing to represent those poses which align well during the dynamics and omitting the rest.

Reviewer #3: Report of Manuscript Number: PONE-D-24-45122

In this study, the authors use docking and molecular dynamics (MD) simulations to investigate the potential application of brazilin and hematein as anti-acne agents. They compare their results with established anti-acne agents, such as KET and TET. Regarding methodology, these methods are commonly used to study such problems. However, there are a couple of concerns regarding the system preparation and the MD methods, which are crucial. Since the work is quite interesting and the results align with the goals of the project, I suggest that the manuscript may be suitable for publication after careful consideration of the following issues.

1) In the abstract, the authors suddenly use abbreviations such as ‘C. sappan L.’ and ‘C. acnes’ without defining them beforehand. It is always better to define abbreviations first, as the authors did for other ligands, such as TET.

2) “Each protein-ligand complex was then placed in a cubic simulation box (dimension of 1x1x1 nm3) and solvated with TIP3P water molecules” This is quite confusing. If the entire box is only 1 nm3 in size and the authors have placed the enzyme within it, the water box would be too small. It is suggested to use a box size of 1 nm3 starting from the protein boundary.

3) What does 0.15 NaCl mean? Are the authors referring to the concentration?

4) I am quite surprised that the authors only performed 1,000 steps for energy minimization. Typically, at least 5,000 steps are recommended. I would also like to see the minimization energy curve to confirm that the system reached the minimum within those 1,000 steps. Moreover, while steepest descent is a powerful algorithm, it is most effective when the system is far from the minimum. As the system approaches the minimum, steepest descent becomes less effective at properly minimizing the system. Therefore, a conjugate gradient algorithm is always recommended after steepest descent to ensure the system reaches a sufficiently localized minimum structure.

5) If the in-house code has already been published, please cite the relevant reference; otherwise, refer to the SI.

6) “This variation in flexibility contributes to the different dynamics and ligand-binding affinities between the two chains.” This statement appears incomplete. Does the author understand the origin of the high flexibility in these new ligands? This is particularly important because the authors’ goal is to conduct a comparative study with an already established ligands.

7) “It is evident that KET and TET undergo significant displacement within the pocket, particularly KET, while the smaller ligands, BRA and HEM, exhibit less mobility.” This is particularly counterintuitive. If a large ligand shows significant displacement within the active site, it implies that the active site is large enough to accommodate the ligand in various orientations. However, this does not necessarily mean that a smaller ligand will be less mobile. Intuitively, the smaller ligand should be even more mobile.

8) “This study serves as a pioneer effort in elucidating the promising capability of HEM to inhibit C. acne lipase.” I would suggest toning down the phrasing a little bit.

6. PLOS authors have the option to publish the peer review history of their article (what does this mean? ). If published, this will include your full peer review and any attached files.

**Do you want your identity to be public for this peer review?** For information about this choice, including consent withdrawal, please see our Privacy Policy .

Reviewer #1: **Yes: ** Vandana Kardam

Reviewer #2: No

Reviewer #3: No

---

## [Author Response · Author response to Decision Letter 1]

3 Jan 2025

Response to reviewers: PONE-D-24-45122

The binding modes of brazilin and hematein from Caesalpinia sappan L. to Cutibacterium acnes lipase: Simulation studies.

PLOS ONE

We thank all the editors and reviewers for feedback and comments. Please see our point-to-point answers.

> We have revised the main text and SI as suggested.

>We have revised the main text and SI as suggested.

"Kasetsart University Research and Development Institute (Grant no. FF(KU)51.67)

Driving Research and Development of Cutting-edge Innovations for ASEAN's Agricultural Leadership by the Office of the National Economic and Social Development Council, and the Office of the Prime Minister "

> We have included the following statement in our revised cover letter. “The funders had no role in study design, data collection and analysis, decision to publish, or preparation of the manuscript.”

"We would like to would like to thank Kasetsart University Research and Development Institute

(Grant no. FF(KU)51.67) and the Office of the National Economic and Social Development

Council, and the Office of the Prime Minister through Kasetsart University under the project

entitled "Driving Research and Development of Cutting-edge Innovations for ASEAN's

Agricultural Leadership" for financial support. We also thank the Kasetsart University HPC

center (Nontri-AI) for computer support."

"Kasetsart University Research and Development Institute (Grant no. FF(KU)51.67)

Driving Research and Development of Cutting-edge Innovations for ASEAN's Agricultural Leadership by the Office of the National Economic and Social Development Council, and the Office of the Prime Minister "

> We have included the above statement in our revised cover letter.

>We have added the captions in the main text as suggested.

> We have reformat the reference list as suggested.

5. Review Comments to the Author

Reviewer #1: In reviewing the manuscript, I have noted several instances of spelling errors that should be addressed before publication. These minor issues, while not significantly affecting the overall content, could impact clarity of the manuscript. A careful proofreading of the document is recommended to correct these errors.

> Thank you for the comment. We have sent to the language editing service for proofreading. The certificate is attached.

Reviewer #2: There are several technical queries with the manuscript mentioned below, which therefore do not allow this manuscript for publishing in its current state.

1. Pg. 13, line 1; default docking protocol; what are those default protocols, it should be mentioned somewhere in the literature (or in the SI).

> We have added the docking details in the “method” section on pg.5, line 129-133 as seen below:

“Molecular docking was performed using GOLD 5.3 software [25] with default parameters for flexible ligand docking to obtain the ligand-lipase complexes as defined within the program. The binding site was defined as all protein residues within 1 nm from the centre of the active site. The ChemPLP and ChemScore fitness functions available in GOLD were used (1).”

2. Is “1x1x1 nm3” the size of the box or padding distance, if it’s the size of the box as mentioned, then mention the padding distance?

> We apologize for this typo and thank the reviewer for pointing this out. The size of a simulation box is 10x10x10 nm3. We have already revised the size in the “method” section on pg. 5, line 135.

3. Whether PBC effects were taken into consideration or not?

> The PBC was applied. We have added the text in the “method” section on pg. 6, line 135 as seen below:

“The Periodic Boundary Condition (PBC) was applied in all directions.”

4. What is the charge of the protein? How many counter ions were added?

> Each chain has a charge of -15 so the charges of a dimer are -30 in total. We have added this information in the “method” section on pg. 5, line 136-137 as seen below:

“A dimer has the total charges of -30, thus 30 Na+ ions were added as counter ions to neutralize the system”

5. At what pH was the protonation state of the residues (specially H) decided?

> We set pH = 7. All histidines are neutral. Their protonation states were assigned by gromacs software. For clarification, the pH used in this work was added in the “method” section on pg. 5, line 136-138 as seen below:

“A dimer has the total charges of -30, thus 30 Na+ ions were added as counter ions to neutralize the system, then 0.15 M NaCl was introduced into the system to mimic physiological conditions (pH 7).”

6. How were the ligands parametrized? Which force field, which tool?

> We have mentioned the topology generation on pg. 5, line 127-129 as seen below:

“The structures of brazilin (BRA), hematein (HEM), ketoconazole (KET), and tetracycline (TET) were retrieved from the PubChem database [21-23]. Ligand topologies were generated using ACPYPE with the AMBER force field [24].”

7. During energy minimizations was the water relaxed before minimizing the entire system?

> Thank you for a comment. When adding water into a box, the short energy minimization was performed for 100steps before adding counter ions and salt solution. Then, the energy minimization was carried out until the maximum force between atoms was < 1,000 kJ/mol�nm (each system required ~ 2000-2500 steps). Also, the 10-ns equilibration was performed where all waters should have enough time to be relaxed.

8. Since the protocol seems incomplete, can the consistent plateau of minimized energies be provided by the authors in the SI, alongside the equilibrated pressures and temperatures as well?

> Thank you for the question. Please see below for the energy, pressure, and temperature in 10-ns equilibration step of all systems.

Figure 1 Energies, Temperature, and Pressure of all system in the equlibration step.

We have also added this figure in SI and comment in the “method” on pg. 6, line 152-153 as seen below:

“After the course of equilibration, all systems became equilibrated as seen in S2 Fig.”

9. Pg. 15, line 5, the authors say that “This observation is further supported by the electrostatic potential shown in Figure 3C.” but no where in the manuscript is mentioned how the ESPs are generated?

> Thank you for a comment. We apologize for this mistake, and we have added the details of ESP calculations in the “method” section on pg.6, line 165-169 as seen below:

“The molecular electrical potential (MEP) surfaces of all ligands were computed by Gaussian 16 package (2) using Density Functional Theory (DFT) with B3LYP6.31G(d, p) basis set. The GaussView 5 was used for MEP visualization (3). The colour range of electrostatic potential surface (ESP) was chosen from red (-0.05 a.u.) to blue (+0.05 a.u.).”

10. Also corroborating the H-bonding observed in dynamics with ESPs does not go well because one is based on force field while the other is quantum mechanical? So, this corroboration must be omitted or must be explained in some other way? Although the authors may take snapshots from the trajectory and perform QM to justify their findings.

> Thank you for the comment. We understand that the hydrogen bonds cannot be directly reflected by ESP since there are many types of electrostatic interactions. We used ESP as a tool to observe the electrostatic properties of each ligands. On pg. 8, we reported that the binding of bulky KET is driven by hydrophobic interactions which can be confirmed by the ESP in Figure 3C. KET shows more non-polar surfaces than other ligands. We have revised the text on pg. 8, line 192-197 as seen below:

“In contrast, the smaller number of hydrogen bonds in KET indicates that hydrogen bonding is not the primary driving force for KET binding (Figure 3B). Seemingly, the binding of KET is driven by hydrophobic interactions. This suggestion is further supported by the electrostatic potential shown in Figure 3C. The bulky KET exhibits increased hydrophobicity compared to the other compounds. These findings suggest that KET binding is primarily driven by non-polar interactions.”

11. Why in the figure 4 the superimposition of each ligand at 60ns, 250ns, 410ns, and 500ns, are shown; instead of representing them at consistent intervals. It is like choosing to represent those poses which align well during the dynamics and omitting the rest.

> We displayed the snapshots at 60ns, 250ns, 410ns, and 500ns because such points of time are the points where there are the changes in the KET-protein hydrogen bonds in Figure 3B (please see below). Other ligands (TET, BRA, and HEM) appear to maintain their hydrogen bonds throughout the course of simulations, whereas there are some points of time that KET loses hydrogen bonds with a protein (see Figure 2 below). Therefore, we selected the points (60ns, 250ns, 410ns, and 500ns) where there are the changes in the interactions of KET-protein as a reference for displaying the snapshots in order to capture all changes.

Figure 2 (Figure 3B in the main text) Hydrogen bonds between protein and each ligand as a function of time.

Reviewer #3: Report of Manuscript Number: PONE-D-24-45122

In this study, the authors use docking and molecular dynamics (MD) simulations to investigate the potential application of brazilin and hematein as anti-acne agents. They compare their results with established anti-acne agents, such as KET and TET. Regarding methodology, these methods are commonly used to study such problems. However, there are a couple of concerns regarding the system preparation and the MD methods, which are crucial. Since the work is quite interesting and the results align with the goals of the project, I suggest that the manuscript may be suitable for publication after careful consideration of the following issues.

1) In the abstract, the authors suddenly use abbreviations such as ‘C. sappan L.’ and ‘C. acnes’ without defining them beforehand. It is always better to define abbreviations first, as the authors did for other ligands, such as TET.

> Thank you for the comment. We have added the full name in the abstract as suggested.

2) “Each protein-ligand complex was then placed in a cubic simulation box (dimension of 1x1x1 nm3) and solvated with TIP3P water molecules” This is quite confusing. If the entire box is only 1 nm3 in size and the authors have placed the enzyme within it, the water box would be too small. It is suggested to use a box size of 1 nm3 starting from the protein boundary.

> We apologize for this typo. The box size is 10x10x10 nm3. We have already revised the box size in the “method” section on pg. 5, line 135.

The protein complex was placed at the centre where its surface is at least 2nm away from the box edge. Thank you very much for this comment.

3) What does 0.15 NaCl mean? Are the authors referring to the concentration?

> We apologize for this typo. This is 0.15 M NaCl. We have already corrected this in the “method” section on pg. 5, line 137.

4) I am quite surprised that the authors only performed 1,000 steps for energy minimization. Typically, at least 5,000 steps are recommended. I would also like to see the minimization energy curve to confirm that the system reached the minimum within those 1,000 steps. Moreover, while steepest descent is a powerful algorithm, it is most effective when the system is far from the minimum. As the system approaches the minimum, steepest descent becomes less effective at properly minimizing the system. Therefore, a conjugate gradient algorithm is always recommended after steepest descent to ensure the system reaches a sufficiently localized minimum structure.

> Thank you for the comment. After checking, we set the energy minimization to run until the maximum force is under 1000 kJ/mol�nm (where the limit of steps = 50,000, but each system required ~2000-2500 steps). It was my mistake that I did not confirm the setting conditions carefully with my MSc student. The energy below can indicate all systems are relaxed in EM step (Figure 3). After the energy minimization, the 10-ns equlibration was performed to equlibrate the system.

Figure 3 Energies obtained from the energy minimization step

The suggestion to use a conjugate gradient algorithm after the steepest descent will be used in our further study.

5) If the in-house code has already been published, please cite the relevant reference; otherwise, refer to the SI.

> The code that we used is just a simple shell script that put all needed gromacs commands together so it is not published anywhere.

6) “This variation in flexibility contributes to the different dynamics and ligand-binding affinities between the two chains.” This statement appears incomplete. Does the author understand the origin of the high flexibility in these new ligands? This is particularly important because the authors’ goal is to conduct a comparative study with an already established ligands.

> Thank you for the comment. This sentence indicates the dynamic differences between 2 lipase subunits. The presence of BRA and HEM seems to enhance the flexibility of lid domain. We also computed the PCA analysis and added the PCA results of the protein motion in Figure S1B in supplementary information (see below). We have revised the text on pg. 7, line 179-182 as seen below:

“In addition

---

## [Editor Report · Decision Letter 1]

21 Jan 2025

The binding modes of brazilin and hematein from Caesalpinia sappan L. to Cutibacterium acnes lipase: Simulation studies.

PONE-D-24-45122R1

Dear Dr. Chairatana,

We’re pleased to inform you that your manuscript has been judged scientifically suitable for publication and will be formally accepted for publication once it meets all outstanding technical requirements.

Kind regards,

Kshatresh Dutta Dubey

Academic Editor

PLOS ONE
---

## [Editor Report · Acceptance letter]

PONE-D-24-45122R1

PLOS ONE

Dear Dr. Chairatana,

I'm pleased to inform you that your manuscript has been deemed suitable for publication in PLOS ONE. Congratulations! Your manuscript is now being handed over to our production team.

Kind regards,

on behalf of

Dr. Kshatresh Dutta Dubey

Academic Editor

PLOS ONE